# Platelets, a Key Cell in Inflammation and Atherosclerosis Progression

**DOI:** 10.3390/cells11061014

**Published:** 2022-03-17

**Authors:** Ricardo Huilcaman, Whitney Venturini, Lucia Fuenzalida, Angel Cayo, Raul Segovia, Claudio Valenzuela, Nelson Brown, Rodrigo Moore-Carrasco

**Affiliations:** 1Faculty of Health Sciences, University of Talca, Talca 3460000, Chile; rhuilcaman@gmail.com (R.H.); whitneyventurini@gmail.com (W.V.); lfuenzalida18@alumnos.utalca.cl (L.F.); acayo@utalca.cl (A.C.); ralemilio@gmail.com (R.S.); 2Facultad de Ciencias de la Salud, Escuela de Tecnología Médica, Universidad Bernardo OHiggins, General Gana 1702, Santiago 8370854, Chile; 3Center for Medical Research, Medical School, University of Talca, Talca 3460000, Chile; cvalenzuela@utalca.cl (C.V.); nbrown@utalca.cl (N.B.)

**Keywords:** atherosclerosis, platelets, inflammation

## Abstract

Platelets play important roles in thrombosis-dependent obstructive cardiovascular diseases. In addition, it has now become evident that platelets also participate in the earliest stages of atherosclerosis, including the genesis of the atherosclerotic lesion. Moreover, while the link between platelet activity and hemostasis has been well established, the role of platelets as modulators of inflammation has only recently been recognized. Thus, through their secretory activities, platelets can chemically attract a diverse repertoire of cells to inflammatory foci. Although monocytes and lymphocytes act as key cells in the progression of an inflammatory event and play a central role in plaque formation and progression, there is also evidence that platelets can traverse the endothelium, and therefore be a direct mediator in the progression of atherosclerotic plaque. This review provides an overview of platelet interactions and regulation in atherosclerosis.

## 1. The Canonical Roles of Platelets

Platelets are structurally, metabolically and functionally complex enucleated cells that result from the fragmentation of megakaryocytes in the bone marrow [1]. While the role of platelets in primary hemostasis, the first line of healing action against vascular injuries, has been well defined, the roles of platelets in immunological processes, cardiovascular disease (CVD) and cancer, among other pathological processes, have only recently emerged [2,3,4].

Platelets, in conjunction with components of the coagulation system, are responsible for the prevention of blood loss from damaged vessels [5]. In the presence of an intact and healthy endothelium, or in the context of a laminar blood flow, circulating platelets remain in a quiescent discoid state near the apical surface of endothelial cells, without forming stable adhesion contacts, a healthy equilibrium in part due to the anti-adhesive properties of quiescent endothelial cells [6]. These antiadhesive properties of endothelial cells, in turn, depend on multiple factors, such as the presence of negatively charged heparin-like glycosaminoglycans and neutral phospholipids, as well as the synthesis and secretion of platelet inhibitors, coagulation inhibitors, and fibrinolysis activators. After injury, platelets are exposed to highly thrombogenic molecules present in the subendothelial tissue, including collagen, von Willebrand factor (VWF), laminin, and thrombospondin, leading to their activation and adhesion to the injured endothelium [7]. At the subcellular level, platelet activation is the result of an increase in the concentration of intracellular calcium, which leads to a reorganization of the actin cytoskeleton and the centralization of platelet granules, with the platelet surface becoming irregular due to the formation of pseudopods. The adhesion process is followed by the release of the content of platelet granules. Through a mechanism of highly regulated exocytosis, the stored bioactive molecules released by platelets can promote the activation and recruitment of other cells [8,9,10]. Two types of granules can be identified: α granules that contain P-selectin, fibronectin, fibrinogen, factor V, factor VIII, platelet factor 4 (PF4 or CXCL4), platelet-derived growth factor (PDGF) and tumor growth factor-α (TNF-α), and δ granules containing adenosine triphosphate (ATP), adenosine diphosphate (ADP), calcium, serotonin, histamine and epinephrine [9,10,11]. The release of calcium and phospholipids from these granules provides a surface for the assembly of various coagulation factors, while the remaining molecules promote a secondary wave of platelet activation [9,10,11,12]. Subsequently, activated platelets synthesize thromboxane A2 (TxA2), which stimulates platelet aggregation and amplifies the activation signals, establishing a positive feedback. The combined actions of TxA2 and ADP enhance platelet aggregation, leading to the formation of the initial platelet plug that temporarily closes the injury. Additionally, in response to ADP, a conformational change in integrin αIIbβ3 (or glycoprotein IIb/IIIa) is induced in the platelet by “inside-out” signaling, shifting this integrin from a closed or low-affinity conformation to an open or high-affinity one. This enables the integrin αIIbβ3 to expose the binding sites to integrin ligands, such as fibrinogen and VWF [13,14], favoring platelet activation and adhesion [15]. These changes are followed by an “outside-in” signaling that is capable of regulating the reassembly of the cytoskeleton, which, in turn, allows the formation of a stable aggregate of platelets and clot retraction [16]. In addition to hemostasis, platelets can also participate in tissue regeneration, being particularly relevant during the first stages of tissue repair [17].

## 2. The Role of Platelets in Inflammation

More than two decades have passed since platelets ceased to be considered a sort of “cellular debris” that prevented blood loss, to become a central cell type involved in inflammatory processes and innate immune responses against microorganisms [18,19,20,21]. From the first publication in 1946 by Houlihan and Copley, demonstrating the adhesion of bacteria to platelets [22], to the article published by Gaertner, F. et al. in 2017 that demonstrated the migratory capacity of platelets and their ability to interact with bacteria and promote an immune response, a multifaceted view of platelets slowly emerged [23]. Recently, it was reported that platelets are capable of phagocytosing microorganisms, thus contributing to the destruction of infectious agents in a manner similar to macrophages [24]. Moreover, we now recognize the key role of platelets in inflammatory processes [24], acting as key orchestrators of inflammatory responses that underly the atherosclerotic process [25]. In the next section, we focus on the involvement of platelets in this pathology.

## 3. The Role of Platelets in Atherosclerosis

The participation of platelets in atherosclerosis was initially described in the context of thrombosis, an event that follows the rupture of an atherosclerotic plaque [25]. In this process, plaque rupture or erosion triggers the release of a wide variety of prothrombotic mediators (e.g., tissue factor, cytokines and matrix proteins), leading to a rapid activation of those platelets that are circulating in close proximity to the atherosclerotic plaque at the time of rupture, and generating a thrombus that, depending on the artery it obstructs, may have fatal consequences (reviewed in [26]). While a thorough description of the factors and mechanisms involved in thrombus formation is beyond the scope of this review, it has been clear that plaque stability is chiefly dependent on histological features [27]. An unstable plaque is formed by a thin fibrous layer (less than 65 µm thick) that encloses an inflammatory infiltrate in which lymphocytes and macrophages predominate, a large lipid nucleus, and relatively few vascular smooth muscle cells (VSMC) [28,29]. In the next section, we will provide evidence pointing to the ability of platelets to influence the stability of the atherosclerotic plaque by modifying the microenvironment and by modulating the function of other cells, including lymphocytes and macrophages. Thus, in addition to platelets’ involvement in the formation of a thrombus after the rupture or erosion of an atherosclerotic plaque, platelets can also influence the stability of the plaque by modulating the microenvironment at the core of the plaque.

By the end of the first decade of the new millennium, it became clear that platelets, in addition to thrombus formation, were also involved in the first stages of atherosclerosis through their ability to bind dysfunctional endothelium and act as a bridge between leukocytes and endothelial cells [20,30,31,32]. This role depends on the complementary interaction between cell adhesion molecules present on the membranes of activated endothelial cells and platelets. When endothelial cells become activated—due to the disruption of blood flow, biochemical imbalances, or metabolic disturbances (e.g., increased levels of modified lipids or hyperglycemia) —, a rapid conformational change of P-selectin occurs on the surface of these cells that increases its affinity for glycoprotein Ib-α (GPIb-α) on the membrane of activated platelets [20,31,33]. As this interaction is reversible, a reinforcement is provided by endothelial P-selectin glycoprotein ligand-1 (PSGL-1). PSGL-1 binds to P-selectin on platelets, allowing the activation of these cells as they roll along the damaged endothelium [14,34]. Additional interactions are implemented to achieve a stable adhesion between platelets and the endothelium, including the binding of integrin αIIbβ3 to fibronectin and fibrinogen/fibrin, the binding of integrin α5β1 to collagen or fibronectin, and the binding of integrin α2β1 to collagen [15], thus interacting with αvβ3 expressed in the activated endothelial lumen [14]. These combined interactions activate a cascade that ends up with the release of several platelet mediators capable of modulating cellular activities in a paracrine fashion, thus accelerating the inflammatory process during atherogenesis [35]. As a consequence, platelets enter a state of hyperactivation in response to inflammation [21]. It is important to note that this hyperreactivity, reflected by a strong platelet response to ADP, can also be caused by the development of resistance to drugs used in antiplatelet treatments such as clopidogrel [36]. This hyperreactivity allows platelets to recruit leukocytes to the subendothelial compartment of vessels [37].

The recruitment of leukocytes is achieved through the interaction of P-selectin expressed on the surface of platelets with its receptor analogue on leukocytes, PSGL-1. PSGL-1, in turn, promotes the activation of integrins β2 (Mac-1 and LFA-1) in leukocytes, which is necessary for the stable and firmer adhesion that favors the secretion of chemokines, such as RANTES (CCL5) [38,39] and PF4 [38] by these cells. These chemokines induce changes in the expression of adhesion molecules in monocytes, allowing them to adhere to activated endothelial cells and favoring their accumulation and the development of atherosclerosis [31,40,41].

It has been observed that the initial interactions between platelets and P-selectin/PSGL-1 on monocytes change position at the time of monocyte transmigration, moving towards the rear of polarized monocytes, and allowing these cells to free themselves from attached platelets [42]. Nonetheless, histological analyses of atheroma plaques obtained from both Apo-E knockout mice and humans have revealed the presence of platelets in the subendothelial compartment of the plaque, raising the possibility that platelets further contribute to atherosclerosis, perhaps by interacting with macrophages present at the lipid core, thus favoring the development of the plaque in a hypercholesterolemic environment [43]. While the ability of platelets to transmigrate into the subendothelial compartment has not yet been fully clarified, Gaertner, F. et al. (2017) documented that platelets have the cell-autonomous capacity of migrate to sites of vascular injury [23].

In summary, the role of platelets in the interaction between endothelial cells and leukocytes is important for the initiation and progression of the atherosclerotic disease. This knowledge underlies the use of blockers of platelet activity as a therapeutic approach to atherosclerosis. Thus, the blockade of cell surface receptors that mediate the early interactions between platelets and endothelial cells (e.g., glycoprotein Ib-α) changes the progression, and impairs the development, of atheromatous plaques in atherosclerotic lesions, and also reduces the inflammatory component at the plaque [32]. There are also strategies to block integrin αIIbβ3 in platelets, inhibiting their recognition by the damaged or activated endothelium [44,45,46]. Furthermore, current developments have led to the design of novel non-RGD peptides that can interfere with the active conformation of αIIbβ3 without exacerbating bleeding or inducing thrombocytopenia [47], which are common consequences of the use of antiplatelet agents. Other approaches include the blockade of this integrin-dependent signaling [48,49] using small molecules that interfere with SH3 sites of Src kinase, which are crucial for binding to β3 chains of integrins. These latter approaches, therefore, interfere with platelets’ outside-in signaling without compromising platelet function in primary hemostasis [50].

### 3.1. Emerging Roles of Platelets in Atherosclerosis

In the past, platelets were generally characterized as cellular elements that, once activated and bound to the endothelium, had no further roles. Far from this traditional view, however, platelets have now been described as immune cells capable of modulating immune responses [51], a fact that has been supported by studies showing that platelets interact and impinge on virtually every cell type found in local inflammatory responses [52].

Recently, new roles in lipid metabolism have been linked to platelets. For instance, the presence of the scavenger receptor CD36 on the surface of platelets has been shown to contribute to their hyperactivity when entering in contact with oxidized low-density lipoprotein (oxLDL) [53]. In addition, platelets can both modulate monocyte differentiation into macrophages and influence the ability of macrophages to accumulate lipids and become foam cells [54]. Moreover, platelets can release a wide variety of molecules that generate an appropriate environment for monocytes to acquire characteristics of myofibroblasts [55], which, in turn, contribute to the development of a type I collagen-rich fibrous cap at the luminal zone of atherosclerotic plaques [56]. As already mentioned, rupture of the cap underlies most thrombotic events [57]. These results show a dual role of monocytes in the development of the atherosclerotic plaque. First, monocytes contribute to the lipid core through the formation of foam cells [54], and second, stimulated by platelets, monocytes acquire new functions and contribute to the cap formation and plaque stability (platelets and smooth muscle cells therefore affect the differentiation of monocytes) [58]. In addition, it has been shown that monocyte-platelet complexes release extracellular vesicles (EV), which have an active proinflammatory role, stimulating the secretion of cytokines in the atherosclerotic plaque [59]. This regulatory role that platelets exert on monocytes and macrophages is not only restricted to cells of myeloid origin; there is evidence that platelets regulate and promote the adhesion of lymphocytes to dysfunctional endothelia and to the extracellular matrix. Adhesion of lymphocytes, especially T lymphocytes and Natural Killer cells, is dependent on platelets and requires the expression of PSGL-1, Mac-1, and CD40L by lymphocytes [60]. As expected, the disruption of platelet–lymphocyte interaction through blockade of the platelet adhesion molecules P-selectin, integrin αIIbβ3, and CD40L, attenuated platelet-dependent lymphocyte deposition [61]. Still, other observations point to new regulatory roles of platelets on other cells of the vascular environment, under both physiological and pathological conditions. Recently, the consequences of the platelet-endothelial cell interaction on the phenotypic transformation of vascular smooth muscle cells from a contractile to a proliferative phenotype was described in a model of diabetes mellitus [62]. It has also been shown that EVs released by platelets are able to induce phenotypic changes, migration, and proliferation in vascular smooth muscle cells, and, additionally, these EVs facilitate the adhesion between smooth muscle cells and monocytes, stimulating the release of inflammatory cytokines, such as IL-6, and thus favoring a proinflammatory environment [63].

These emerging functions of platelets likely involve the release of the contents of their intracellular granules [64]. One of the most studied mediators is platelet factor 4 (PF4), a molecule with the ability to attractant monocytes, neutrophils and fibroblasts [30]. PF4 also favors platelet activation and migration, and enhances the uptake of oxidized LDL by macrophages, promoting the formation of foam cells and contributing to the development of the lipid core of atherosclerotic plaques [65,66]. In the case of vascular smooth muscle cells, PF4 induces an inflammatory secretory phenotype partly by activation of the transcription factor KLF4 [67] and also stimulates the proliferation and calcifying potential of VMSCs [68]. Another factor that has recently been found in the secretory granules of platelets is migration inhibitory factor (MIF), a chemokine also produced by macrophages and endothelial cells. This molecule has been linked to adhesion and transmigration of monocytes into the atherosclerotic lesion [68]. CD40 is also an important factor released by platelets that has been shown to be important in inflammatory and thrombotic processes [69]. Once bound to the membrane of activated platelets, this protein can be cleaved, and produce transactivation of both platelets and endothelium and leukocytes by CD40L [70] generating vascular immune responses [71].

On the other hand, there are numerous lipid mediators present in platelets that actively participate in signaling and intercellular communication [72]. An example of these mediators is platelet-activating factor (PAF or 1-O-alkyl-2-acetyl-sn-glycero-3-phosphorylcholine), this phospholipid is produced in different types of cells after stimulation, among the cells that produce this mediator are: platelets, endothelial cells, monocytes, among others. After its production and release, it binds to its PAF receptor (PAFR) and triggers a signaling cascade that, in the case of platelets, produces a strong activation [73]. The release of PAF is associated with the process of platelet secretion after activation and aggregation. It has been shown that thrombin and other agonists can stimulate the release of PAF through platelet-derived microparticles (PMPs) [74]. This important fact helps us to understand how platelets through of the release of PMPs actively participate in inflammatory processes. PAF and other oxidized phospholipids can bind to this receptor and trigger a powerful inflammatory response, which must be regulated to maintain homeostasis. An important role in the regulation of the biological activity of PAF and other oxidized phospholipids is played by PAF-acetylhydrolase (PAF-AH or lipoprotein associated phospholipase A2), a calcium-independent phospholipase that degrades PAF and other mediators into inactive metabolites [75]. In humans, PAF-AH circulates through the bloodstream associated with lipoparticles, particularly low-density lipoparticles (LDL) and, to a lesser extent, in high-density lipoparticles (HDL) [76]. In 2006, Mitsios et al. demonstrated that this enzyme is secreted by platelets through PMPs [77]; this enzyme is the focus of intense research given its importance in the regulation of inflammation in sepsis. In addition to the above, it has been shown that PAF, anchored to the plasma membrane of activated endothelial cells and adherent platelets, participates in the adhesion, rolling and subsequent extravasation of polymorphonuclear cells (PMN) [78,79]. Undoubtedly, PAF is a link between platelets, endothelial cells, and leukocytes in the context of inflammation and atherosclerosis [80].

### 3.2. Platelet Recruitment into the Atherosclerotic Lesion and Transendothelial Migration

It has been suggested that platelets have the necessary machinery to execute cell-autonomous transendothelial migration. First, there is evidence that platelets can migrate attracted by cytokines [81] and chemokines [82]. There is also indirect evidence that supports transendothelial migration by platelets. These studies, which were carried out in both human tissues and tissues derived from ApoE-KO mice, demonstrated the presence of platelets within atherosclerotic plaques, that is, platelets readily detectable in the intima of the affected blood vessels and in close proximity to tissue macrophages [43]. In another study, in which a stroke was induced in mice by a transient middle cerebral artery occlusion, the presence of platelets outside the vasculature was observed in ischemic mouse brains [83]. Local cerebral ischemia activates platelets by promoting FasL expression on the platelet surface, thus promoting apoptosis in brain tissue directly [83].

So far, however, the mechanism involved in the translocation of platelets to the subendothelial space, and the contribution of these platelets to the development of the atheromatous plaque, have not been fully clarified. In 2012, Van Lammeren’s group demonstrated, through histological analyses of 188 atheroma plaques isolated from carotid arteries, that platelets can be found within the plaque in association with signs of vessel rupture or small hemorrhages [84]. These findings raised the possibility that platelets were mere markers of small ruptures of the internal vessels that supply the atherosclerotic plaque, which, in turn, would have an important prognostic value in patients with atherosclerosis [84].

Nonetheless, it has now become accepted that platelets are capable of not only gaining access to the plaque through micro-ruptures of vessels [84], but of actively and cell-autonomously migrate through the endothelium towards the plaque core [85]. Based on these migratory properties, a recent study has proposed the use of platelets as markers of atherosclerotic plaque formation [86]. In particular, the authors of this work proposed the use of nanoparticles fused to antiplatelet antibodies for an NMR-based detection method of atherosclerotic plaques in vivo, offering new approaches for the study of atherosclerosis [86].

Platelets may also acquire the ability to migrate through the epithelial barrier of the intestine, either bound or in close proximity to neutrophils. Migrating platelets in this case may contribute to the development of intestinal inflammatory processes by regulating the activity of lymphocytes [87]. It has been suggested that the presence of platelets at the intestinal lumen can be the result of a carry-over mechanism conveyed by neutrophils, although there may be a coordination between both types of cells for the mobilization of platelets through cell monolayers. Although these results could be interpreted as distinctly different from the results obtained in models of atherosclerosis, they do highlight the ability of platelets to migrate through cellular barriers. In support of this, Massberg’s group evaluated the ability of platelets to adhere to sites of vascular injury and inflammation, showing that platelets not only are the first cells to reach the sites of injury, but also have the ability to migrate in search of bacteria that are subsequently targeted for phagocytosis [23]. This constitutes irrefutable evidence of the migratory capacity of platelets.

Other studies, carried out in models of cardiovascular diseases, unveiled yet another mechanism that allows platelets to migrate into atherosclerotic lesions. In this case, platelets, previously stimulated with oxLDL, are first phagocytosed by monocytes and carried passively to the atheroma core [66]. This is consistent with the finding reported by Gonzalez et al. (2014), in which platelets could be detected in the intima of arteries, both as elements phagocytosed by macrophages and as free platelets in an atherosclerotic plaque [43]. Taking advantage of an in vitro model that recapitulates the conditions that lead to the development of atherosclerosis, we demonstrated that platelets are capable of migrating through an intact endothelium. Interestingly, for this event to occur, platelets require signals provided by circulating monocytes [88]. In this model, monocytes and platelets were first stimulated by inflammatory (TNF-α) and metabolic (LDLox) mediators before testing transendothelial migration. Our results demonstrated that platelets can migrate through a monolayer of activated endothelial cells. Interestingly, when pretreated with conditioned medium derived from cultures of monocytes, platelets migrated at a rate similar to that observed when platelets were co-cultured with monocytes. This observation suggests the existence of factors secreted by monocytes that stimulate platelet migration [85,88].

## 4. Concluding Remarks

Platelets have proved important in the development of atherosclerosis, in part due to their ability to initiate monocyte migration at sites of vascular inflammation. In addition, new capabilities, such as their cell-autonomous ability to migrate through the endothelium and their ability to regulate inflammatory activities, are now becoming critical aspects in the progression of atherosclerotic lesions. Therefore, finding ways to modulate platelets’ functions through manipulation of their signaling machinery is presently a focus of great biomedical interest, opening the possibility to develop therapies using, for instance, naturally occurring molecules.

At sites of inflammation, platelets act as a bridge between the endothelium and monocytes. This event facilitates the adhesion and migration of leukocytes to the atheromatous plaque. It is also known that platelets can induce the transformation of monocytes to macrophages, and thus influence the formation of foam cells (Figure 1). In addition, platelets can release mediators that promotes a proinflammatory response with more invasion and proliferation of cells in the vascular intima. Platelets also could promote LDL oxidation. Therefore, far beyond their role in hemostasis, platelets are recognized as immune components.

## Figures and Tables

**Figure 1 cells-11-01014-f001:**
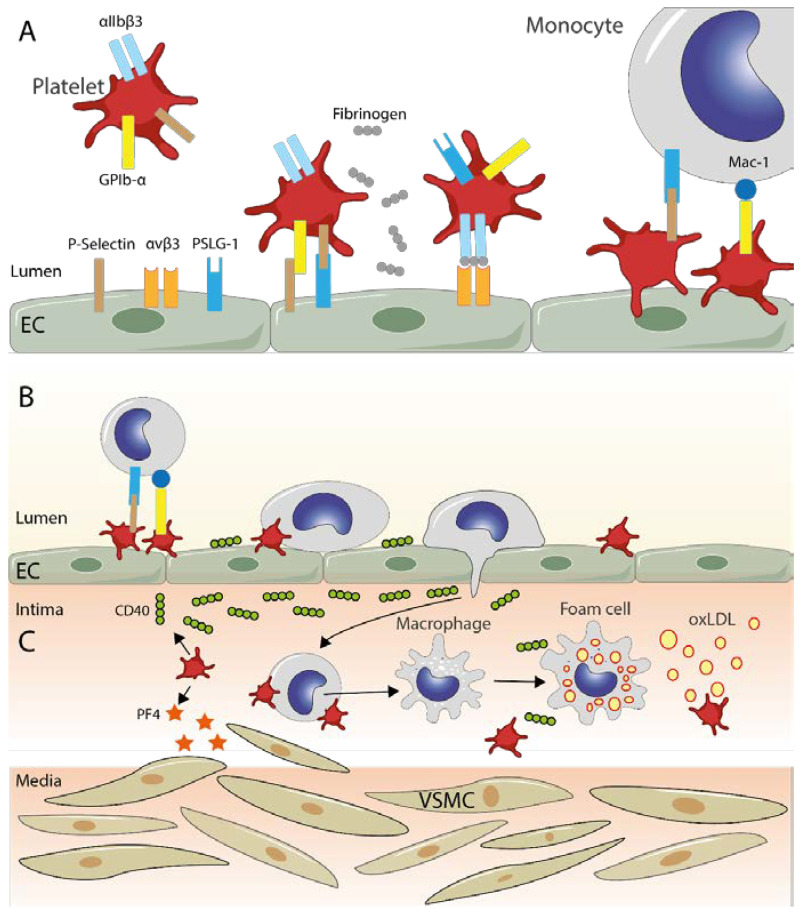
Transendothelial migration of monocytes and platelets. (**A**) Turbulent blood flows or pro-atherogenic metabolic conditions, among other factors, trigger the expression of adhesion molecules by activated endothelial cells, ultimately leading to platelet adhesion and activation. First, there is a binding between endothelial P-selectin and platelet GPIb-α, which is reinforced by the binding of P-selectin and P-selectin glycoprotein ligand (PSLG-1), expressed on activated platelets and endothelial cells, respectively. A more stable binding, mediated by fibronectin (gray chained circles), is established between integrin αvβ3 of endothelial cells and αIIbβ3 of platelets. Once platelets have attached to the activated endothelium, they can function as a bridge between monocytes and endothelial cells thanks to the interactions between PSLG-1 and P-selectin, and between Mac-1 and GPIb-α, promoting the migration of monocytes into the intima (**B**). (**C**) Platelets can also modulate the composition of the intimal layer of blood vessels by releasing CD40 (green chained circles), which favors increased migration of monocytes and their transformation into foam cells by the inclusion of oxidated LDL (yellow circles). As shown, platelets can release Platelet Factor 4 (PF4, orange stars), which, among other effects, allows the recruitment of vascular smooth muscle cells (VSMC), favoring their proliferation and change to a proinflammatory phenotype.

## Data Availability

Not applicable. No data were generated or analyzed in this work.

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
