# Peer review of "Platelets, a Key Cell in Inflammation and Atherosclerosis Progression"

_cells, 2022, doi:10.3390/cells11061014_

Round 1

Reviewer 1 Report

Huilcaman et al present a short and concise overview on the role of platelets in inflammation and atherosclerosis. A short description or a paragraph describing the transition to thrombosis would be useful

Other minor points:

  • Seems that some references are repeated (36 and 39), and some are not included (page 4, 5)
  • For consistency, please change GPIIbIIIa into alphaIIbbeta3 (as in other places in the text and in the figure).
  • Can authors describe (mention) in the main text the binding of the P-selectin on the endothelial cell to PSGL-1 on activated platelets (as it is depicted in Figure). Is there binding of P-selectin from activated platelet to endothelial PSGL-1?
  • page 4, line 152 “These results show a dual role of monocytes in the development of atherosclerotic plaque, on the one hand...” – language correction and insert reference
  • page 4, line 175 “Furthermore human platelets…” – insert original reference on LRP8
  • page 4, line 183 “This molecule has been…” - insert reference on MIF
  • page 5, line 236-240 “Interestingly, when platelets were stimulated with a medium that had been conditioned…” – the sentence is not clear, please change or rephrase.
  • Figure 1 – in figure legend in A) more details are needed, including all mentioned molecules and their full names. Drugs that block GPIIbIIIa in the figure – not clear why are here since not mentioned or explained, namely in the main text. In B) mention which adhesion molecules.
  • Incorporate Fig. 1 in the main text (it is not mentioned)

Author Response

Dear Reviewer 1,

Thank you very much for your comments, they have allowed us to significantly improve our work.

We have incorporated a paragraph that briefly explains the transition to thrombosis, focusing the story on the role of the platelet in this process. However, we know quite well the mechanisms associated with platelet activation once the plaque has been eroded or ruptured, we know little about the factors that influence whether a plaque is stable or unstable, where platelets will probably also play a central role. We still have much to learn from this process.

On the other hand, all minor comments were incorporated or corrected in the text, and we have also improved the figure based on your suggestions.

Reviewer 2 Report

In this review manuscript by Huilcamán and colleagues, the different roles of platelets in inflammation and atherosclerosis are outlined. The topic of the overview is interesting, and the functions of platelets beyond hemostasis are worth communicating to the broad scientific community. The manuscript is clearly written but needs a number of improvements and clarifications.

The most important issue is the use of references. In many cases, the authors refer to review studies when discussing key mechanisms, where the original work should be cited. In addition, some citations could be more accurate.

Specific comments:

Line 50 : platelet factor 4 (CXCL4). in general, please also add the systematic names when mentioning chemokines (e.g. RANTES: CCL5).

Line 61: please be consistent when referring to GPIIb/IIIa (αIIbβ3)

Line 62: please be consistent with abbreviations (vWF vs vWf)

Line 79: The work by Oggero and colleagues (ref 24) is interesting enough to discuss in detail. And also other studies that have investigated platelet extracellular vesicles could be discussed. At the position of reference 24, a general overview should rather be cited (e.g. ref. 19 or Pubmed ID: 33468314)

Line 82: “plaque [25].”

Line 102: “hyperactivation”, platelet hyperreactivity, either in the form of resistance to drugs (e.g. PMID: 30091133), or by a lower activation threshold also accelerates atherosclerosis (PMID: 25472975). This should be discussed.

Line 109: The authors refer to the chemokines CCL5 and CXCL4, but the reference (19) is not entirely correct. Original work on these platelet chemokines is e.g. PMID: 11282909, reference 26 and PMID: 19122657). Also references 27 and 33 are review papers.

Line 132: blocking platelet αIIbβ3 blocks platelet aggregation, but this reviewer doubts that activation is also inhibited (outside-in activation at most). With respect to this section (line 131-136), there is an interesting paper that may be discussed (PMID: 35023301).

Line 144: The reviewer has read the paper of Yeaman (ref 42) in the past, but cannot remember that there was much mention of a role of platelets in atherosclerosis. Perhaps a different reference is more appropriate.

Line 152: the concept of (fibrous) cap should be introduced better.

Line 166 (section): The authors might also discuss a study on the effects of platelet extracellular vesicles on a (pro-atherogenic) phenotype of smooth muscle cells (PMID: 28717419).

Line 170: the section on platelet factor 4 can benefit from a mention of other relevant studies (e.g. 23568488, 35054772).

Line 184: I think that the authors mean to cite the title between “ “ here.

Line 189: I wish to point to a further study where platelets were found in the tissue during inflammation (PMID: 26232171).

Line 204: Please cite the original work here.

Figure 1: Perhaps the authors can include the following processes in the figure:

- deposition of chemokines and other factors on endothelium.

- surface expression of cytokines e.g. CD40 and CD40L

- entry of platelets via plaque microvessels

Author Response

Dear Reviewer 2,

Thank you very much for your comments and observations, they have allowed us to significantly improve our work.

Regarding the references, we have incorporated the original works in most cases, leaving only the revisions that are key to understanding the mechanisms. In addition, we have incorporated new references to original works that help to better understand the processes and mechanisms involved.

Regarding your specific comments, we have corrected and incorporated most of your comments, which has greatly improved the understanding of the work.

Finally, we have included in the figure the deposit of chemokines, the surface expression of CD40, among other improvements to the figure.

Reviewer 3 Report

This review is very interesting and offer the current view on the field. It should be published, although I found the inflammatory part a bit reduce compare to atherosclerosis part.

Authors should better address more clearly what is consistent with inflammatory to what is typical of atherosclerosis.

Repeat in lane 115 "of of"

Author Response

Dear Reviewer 3,

Thank you very much for your comments and observations, they have allowed us to significantly improve our work.

We have partially modified the way events are reported in order to give greater relevance to the inflammatory process in atherosclerosis. In addition, we have corrected typos and repeated words.